# Genome-Wide Identification and Functional Characterization Reveals the Pivotal Roles of *BnaA8.ATG8F* in Salt Stress Tolerance and Nitrogen Limitation Adaptation in Allotetraploid Rapeseed

**DOI:** 10.3390/ijms231911318

**Published:** 2022-09-26

**Authors:** Tianyu Zhang, Ting Zhou, Yifan Zhang, Junfan Chen, Haili Song, Pengjia Wu, Caipeng Yue, Jinyong Huang, Zhenhua Zhang, Yingpeng Hua

**Affiliations:** 1School of Agricultural Sciences, Zhengzhou University, Zhengzhou 450001, China; 2School of Life Sciences, Zhengzhou University, Zhengzhou 450001, China; 3College of Resources and Environment, Hunan Agricultural University, Changsha 430128, China

**Keywords:** autophagy, *Brassica napus*, *BnaA8.ATG8F*, multiomics analysis, N starvation, salt stress

## Abstract

Autophagy is a common physiological process in organisms, including higher plants. The *ATG8* subfamily, the core member of the autophagy-related gene (ATG) family, plays a key role in plant growth and development and nutrient stress responses. However, the core *ATG8* homologs and their roles in stress resistance remain elusive in allotetraploid rapeseed (AACC, *Brassica napus* L.). In this study, we identified 29 *ATG8* subgroup members, consisting of three phylogenetic clades, based on the analysis of genomic annotation and conserved motifs. Differential transcriptional responses of *BnaATG8s* to salt stress, nitrogen limitation, and other nutrient stresses were investigated, and we identified *BnaA8.ATG8F* as the core *ATG8* member through gene co-expression network analysis. Decreased *BnaA8.ATG8F* expression repressed the salt tolerance of transgenic rapeseed plants by significantly reducing the root Na^+^ retention under salt stress. Moreover, downregulation of *BnaA8.ATG8F* increased nitrogen (N) limitation sensitivity of transgenic rapeseed plants through decreasing N uptake, translocation, and enhancing N remobilization under nitrogen starvation. In summary, we identified the core *ATG8* homologs and characterized their physiological and molecular mechanisms underlying salt stress tolerance and nitrogen limitation adaptation. Our results may provide elite genetic resources for the genetic improvement of nutrient stress tolerance in rapeseed.

## 1. Introduction

In order to survive in a variety of adverse environments, plants have evolved various strategies to resist or adjust to stresses. Autophagy, an evolutionarily conserved self-eating mechanism among various organisms, refers to recycling damaged proteins and specific compounds and activates some stress-response pathways [1,2]. As one of the mechanisms of self-defense during the plant evolution process, autophagy can be induced by nutrient starvation and is vital for self-digestion to occur in lysosomes or vacuoles of eukaryotic cells [3]. Autophagy-related genes (*ATGs*) were first discovered in yeast, and this physiological process has attracted extensive attention since then [4]. So far, a total of 35 *ATGs* have been identified in yeast, and almost all of these genes have close homologs in plants [5]. In higher plants, autophagy plays an important role in many biological processes, such as starvation responses [6], oxidative stress [7], drought stress, and pathogen infection. Autophagic mutants or autophagic gene silencing plants were sensitive to environmental stresses. On the contrary, the resistance of autophagy overexpressed plants against various stresses was enhanced [8,9].

According to previous studies, the core process of autophagy in plants is composed of 18 proteins, which belong to several parts of the protein complex [10]. First, when cells are under normal conditions, the phosphorylation of the target of rapamycin (TOR) kinase-dependent ATG1/ATG13 regulatory complex inhibits the function of phagophores. However, under stress conditions, the TOR activity is inactivated by stress signals, which induce the formation of the pre-autophagosomal structure in the participants of the ATG1-ATG13 complex [11]. Subsequently, the phosphatidylinositol 3-kinase complexes, including VPS15, VPS34, VPS38, and ATG6, modify phagosomes with phosphatidylinositol-3-phosphate (PI3P) to recruit ATG18, ATG2, and ATG9 complexes for membrane elongation [12]. The ATG8/ATG12 ubiquitin-like conjugation system mediates phagosome expansion and autophagosome maturation [13].

Among the two ubiquitin-binding systems, *ATG8* seems to play a more central role in plant autophagy. Many studies have shown that the autophagy process involving *ATG8* can affect plant responses to nutrient stresses. In *Arabidopsis thaliana*, overexpression of *ATG8* can stimulate autophagy activity and improve nitrogen transport efficiency [14]. In addition, t *ATG8* is necessary for cell survival in the absence of nitrogen, phosphorus, and sulfur in *Chlamydomonas* [15]. In most plant genes encoding core mechanisms of autophagy, knockout mutants are more sensitive to carbon starvation and nitrogen deficiency than wild-type plants and, in some cases, have accelerated properties [16]. However, little is known about the identification of the core *ATG8* gene and its function in response to nutrient stresses in oilseed rape.

Oilseed rape (*Brassica napus* L.) is a crucial oil crop worldwide. The allotetraploid *B. napus* (A_n_A_n_C_n_C_n_, 2n = 4× = 38) is formed by natural distant hybridization between diploid *B. rapa* (A_r_A_r_, 2n = 2× = 20) [17] and diploid *B. oleracea* (C_o_C_o_, 2n =2× = 18) [18], with complex origin and diversity [19]. *B**. napus* has higher nutrient requirements than other crops, whereas it is easily affected by stresses, such as salt stress, various nutrient stress, heavy metal stress, and other abiotic stresses during vegetative growth, especially at the seedling stage. However, systematic studies on autophagy-related gene families are still limited due to the complexity of the rapeseed genome.

In this study, we used the ‘Westar’ as the experimental species because of its good transgenic efficiency and complete genome annotation [20]. The main purpose of this study was as follows: (i) genome-wide identification of *ATG8* homologs in rapeseed, (ii) molecular characterization of transcriptional responses and core members of *ATG8s* to various nutrient stress, and (iii) functional dissection of the roles of the core *ATG8* member under salt stress and nitrogen limitation. Our results may provide elite genetic resources for the genetic improvement of nutrient stress tolerance in rapeseed.

## 2. Results

### 2.1. Genome-Wide Identification of ATG8s in Brassica Species

The *Arabidopsis* genome contains nine *ATG8* genes including *AtATG8A*, *AtATG8B*, *AtATG8C*, *AtATG8D*, *AtATG8E*, *AtATG8F*, *AtATG8G*, *AtATG8H*, and *AtATG8I*. In order to further compare the evolutionary diversity of *ATG8* genes among *Brassica* species, *B. rapa*, *B. oleracea*, and *B. napus* were identified by the method mentioned below. The results showed that 12, 17, and 29 members were identified in *B. rapa*, *B. oleracea*, and *B. napus*, respectively (Appendix A). It is generally believed that copy number variations of *ATG8s* in plant species may be the result of evolutionary selection. Although we found that the number of *ATG8* genes in *B. napus* was very close to those in *B. rapa* and *B. oleracea*, gene loss and gene replication still occurred in some members. Specifically, *BnaATG8A*, with a total of eight members, had the most members in the *BnaATG8* gene family. Interestingly, no homologs of *ATG8B* and *ATG8G* were found in the rapeseed genome, and *ATG8G* was not even identified in *Brassica* crops (Appendix A). This might suggest that the gene was lost before *Brassica* species formed because of functional redundancy.

### 2.2. Genomic Distribution, Expansion, and Syntenic Analysis of BnaATG8s

According to the genomic annotation from the *B**. napus* pan-genome database, we found a total of 29 *BnaATG8s* distributed on 16 chromosomes. Among them, 15 *BnaATG8s* are located on eight chromosomes (A1, A2, A3, A4, A5, A7, A8, and A9) of the A subgenome, and the remaining 14 genes are located on eight chromosomes (C1, C2, C3, C4, C5, C6, C8, and C9) of the C subgenome (Appendix A). 

Subsequently, the collinearity analysis was performed on rapeseed with *Arabidopsis*, *B. rapa,* and *B. oleracea*, respectively. Specifically, a total of 24 (82.8%) *BnaATG8s* were highlighted in collinearity between *Arabidopsis* and *B. napus* (Figure 1A). There were 13 (86.7%) *ATG8s* of the A subgenome and 12 (85.7%) of the C subgenome (Figure 1A). 

Gene amplification is a main driving force of adaptive evolution [21]. To understand the expansion patterns of the *BnaATG8* family genes in *B. napus*, we investigated gene duplication events based on the physical location and gene collinearity of *BnaATG8s* (Figure 1A). The *Arabidopsis* genome can be divided into 24 ancestral Cruciferous segments labeled A–X [22]. Based on a comparative analysis of *A**. thaliana*, the 19 chromosomes of *B**. napus* were also defined as 24 blocks [23]. *AtATG8* and its corresponding homologs in *B**. napus* are located on the same chromosome block. Specifically, *ATG8s* are located in seven chromosome blocks, including U, O, D, G, J, N, and F (Appendix A). In conclusion, during the formation and evolution of *B. napus*, most members of the *ATG8* gene family remained intact except *ATG8B* and *ATG8G* in *B. napus*.

### 2.3. Phylogenetic Relationships and Evolutionary Selection Pressure of the ATG8 Proteins

To further reveal the molecular and phylogenetic relationships of the *ATG8* gene family between *Brassica* crops and *Arabidopsis*, a rootless phylogenetic tree was constructed using the protein sequences of BnaATG8s, BolATG8s, BraATG8s, and ATATG8s (Figure 1B). In *Arabidopsis*, the ATG8 family members were mainly classified into three clades: Clade I (AtATG8A and AtATG8B), Clade II (AtATG8C, AtATG8D, AtATG8E, AtATG8F, and AtATG8G), and Clade III (AtATG8H and AtATG8I) (Figure 2). In each evolutionary clade, the members of *Brassica* ATG8 cluster with their corresponding homologs in *Arabidopsis*. The results showed that *ATG8s* had diverged before the formation of *Brassica* species.

To describe selection pressures for the ATG8 proteins during evolution, we used homologous *ATG8* gene pairs between *B. napus* and *Arabidopsis* genomes to estimate Ka, Ks, and Ka/Ks (Appendix A). According to Darwinian evolution theory, it is generally accepted that if Ka/Ks > 1.0, positive selection has occurred. Ka/Ks < 1.0 indicates that purifying selection has occurred, and Ka/Ks = 1.0 refers to neutral selection [24]. The results showed that the Ka/Ks value of the *BnaATG8* gene family was much less than 1.0. Therefore, we infer that ATG8s might undergo very strong purification selection to maintain gene structure and function. Subsequently, we used the following formula to estimate the divergence time of *BnaATG8s* from their progenitors: T = Ks/2λ, λ = 1.5 × 10^−8^ for *Brassica* species [25]. The separation of *Arabidopsis* and *Brassica* species occurred 12–20 million years ago (Mya) [26]. The results showed that the possible difference between *BnaATG8s* and *AtATG8s* was about 8–18 Mya, with an average of 13.97 Mya, suggesting that speciation was accompanied by the divergency of *ATG8**s*.

### 2.4. Molecular Characterization of BnaATG8s

To further understand the molecular characteristics of the *BnaATG8* gene family, we used ExPASy to calculate the physicochemical properties of each member. In general, CDS lengths of *BnaATG8s* ranged from 201 bp (*BnaA3.ATG8C*) to 861 bp (*BnaC8.ATG8F*), and the corresponding derived amino acid numbers ranged from 66 to 286 (Appendix A). The molecular weight of BnaATG8s protein varied from 7.70 kD to 33.42 kD, which is directly related to the number of amino acids (Appendix A). The theoretical isoelectric point (pIs) of BnaATG8s varied from 5.2 (BnaC8.ATG8F) to 9.45 (BnaA3.ATG8C). The grand average of hydropathy (GRAVY) value is defined as the sum of hydropathy values of the amino acids divided by the protein length. The results showed that BnaATG8s had GRAVY values ranging from −0.544 (BnaC4.ATG8E) to 0.046 (BnaC6.ATG8F) (Appendix A). If GRAVY < 0, then we consider the protein to be hydrophilic. Therefore, all members except BnaC6.ATG8F is predicted to be hydrophilic. BnaATG8s had instability indices ranging from 26.79 (BnaA7.ATG8E) to 50.46 (BnaA3.ATG8D), which indicated that there are great differences in protein stability in the BnaATG8 family (Appendix A).

WoLF PSORT was used to predict the subcellular localization of 9 AtATG8s and 29 BnaATG8s. The results indicated that, except for AtATG8E and AtATG8H and their homologs in rapeseed which were located in the nucleus, the remaining proteins were predicted to be located in the cytoplasm (Appendix A).

Phosphorylation plays a crucial role in signal transduction as an epigenetic regulation mechanism of protein activity, which usually occurs at the sites of serine, threonine, and tyrosine [27]. The results showed that the BnaATG8 proteins have different phosphorylation preferences for these amino acid sites, whereas most phosphorylation sites of BnaATG8s are serine (Appendix A).

The transmembrane domain (TM), which is usually represented by the effector region of transmembrane proteins, is the main binding site of proteins and lipids. According to the predicted results, only BnaC6.ATG8F has one transmembrane domain, and the rest proteins are not predicted to have transmembrane domains (Appendix A). Therefore, we suggested that members of the BnaATG8 family may function in the cytoplasm or nucleus rather than the cell membrane.

### 2.5. Conserved Domains, Gene Structure, Protein Interaction, and Transcriptional Regulatory Analysis

In general, if the residues of amino acids are evolutionarily conserved, they are considered to be relatively important for structure and function. Based on the previously mentioned identification of BnaATG8 members using PFAM, these BnaATG8 proteins all contain only one ATG8 domain. The MEME was used to predict conserved motifs of 9 ATATG8 proteins and 29 BnaATG8 proteins, and five motifs were identified based on MEME analysis. Three of the predicted motifs (Motif I, Motif II, Motif Ⅲ) were identified as domains of ATG8 according to the Pfam database annotation. Notably, as shown in Motif I, a conserved glycine was identified at the C-terminal of the protein, which is typical of the ATG8 family [28] (Figure 1D). Each member of the BnaATG8 protein contains at least two of these three motifs except BnaA3.ATG8C, which contains only the Motif II (Figure 1C). These results indicated that the functional divergency of BnaATG8 proteins might be due to motif diversity.

Subsequently, we determined the exon-intron structure of *BnaATG8s* based on the genome annotation information of rapeseed (‘Westar’) (Appendix A). The statistical results showed that the number of exons in the *BnaATG8* family ranged from three (*BnaA3.ATG8C*) to 8 (*BnaC8.ATG8F*), and most members have five exons (Appendix A, Appendix A). In general, the diversity of exon-intron structures reflects the evolution of gene families. The exon number changes might be caused by alternative splicing, which affected the functional segregation among *BnaATG8s*.

To further identify which proteins are likely to interact with BnaATG8s, we constructed a protein interaction network using the STRING database based on known experiments and text mining to predict likely interactions (Appendix A). According to the results of predicted interaction networks, we found that almost all members of ATG8 could interact with the ATG4A/B (AT2G44140/AT3G59950) protein, which was consistent with previous studies. In *A**. thaliana*, ATG4s can cleave the C-terminus of ATG8, and the double mutant of *atg4a/atg4b* shows autophagy defects [29]. In addition, ATG3 and ATG7 have been found to interact with ATG8s, which has also been confirmed in other species [29].

Transcription factors (TFs), as upstream regulatory genes, play a core role in transcriptional regulation and can combine with *cis*-acting regulatory elements (CREs) in the promoter region of downstream target genes. In this study, the 1.5-kb genome sequences upstream of the start codon (ATG) of *BnaATG8s* were extracted for CRE prediction. The predicted results indicated that the transcriptional response of *BnaATG8s* might be regulated by multiple networks (Figure 2). In addition to filtering out the common promoter region elements such as the TATA-box, CAAT-box, and unknown elements with unannotated functions, we also identified several phytohormone responsive elements, including the salicylic acid-responsive (TCA), ABA-responsive (ABRE; ACGTG), and auxin-responsive elements (TGA; AACGAC) and some elements involved in the MeJA response. Notably, a large number of TGACG-motifs were also identified in the promoter regions. The following over-presented CREs were further concerned, including CREs required for anaerobic induction, recognition, and binding sites of MYB and MYC, some of which have proved to be involved in the response of plants to abiotic stresses (Figure 2).

### 2.6. Transcriptional Analysis of BnaATG8s under Diverse Nutrient Stresses

We first predicted the transcriptional pattern of *AtATG8s* in *A**. thaliana* using the TAIR database. The results showed that almost all of the nine *ATG8s* in *A**. thaliana* were highly expressed in cytoplasmic matrix and vacuole (Appendix A). In order to further understand the biological functions of *BnaATG8s* in response to various nutrient stresses, we analyzed the transcriptional responses of rapeseed plants to nitrate deficiency, ammonium toxicity, phosphorus deficiency, potassium deficiency, salt stress, and cadmium toxicity.

Salt stress, as a common agricultural problem in rapeseed, is an important factor restricting crop growth and development. To further explore the role of *BnaATG8s* in response to salt stress, we analyzed their transcriptional profiling under salt stress. The results showed that a total of 13 *BnaATG8* genes were identified as differential expressed genes (DEGs) in the shoots, all of which were up-regulated. Among them, *BnaC3.ATG8A*, *BnaA8.ATG8F**,* and *BnaC8.ATG8F* have higher gene expression and differential expression multiple, which might indicate that they play a core role in salt stress resistance as downstream functional genes. Three DEGs (*BnaA3.ATG8E*, *BnaA7.ATG8E*, and *BnaC3.ATG8E*) were found in the roots, and all of them showed a downward expression trend after salt treatment (Figure 3A). Among all the DEGs, only three *BnaATG8E* subfamily members were down-regulated in the roots under salt stress, while the expression levels of the other members were up-regulated in the shoots but not significantly differentially expressed in the roots. This indicates that *BnaATG8Es* might have a different expression pattern from other gene family members.

Under nitrate deficiency, a total of 15 *BnaATG8s* were differentially expressed in the shoots, whereas only three of 15 DEGs were also identified in the roots (Figure 3B). Interestingly, we found that all the DEGs were significantly up-regulated under low nitrogen treatment. *BnaATG8F* was the most obviously up-regulated gene with the highest expression level and differential expression multiple (Figure 3B). In addition, we found that *BnaATG8Cs* and *BnaATG8Ds* members were not significantly differentially expressed, which might indicate that these genes had limited functions in rapeseed response to low nitrate stress.

To further understand the role of autophagy in response to low potassium stress, we analyzed the transcriptional response of the *BnaATG8* gene family under low potassium conditions. A total of seven *BnaATG8s* were identified as the DEGs, and all of them were up-regulated in the shoots under potassium deficiency. Only *BnaC3.ATG8A* showed a significant difference in the root expression (Figure 3C).

Phosphate plays an important role in plant photosynthesis, growth, and development. To further understand the role of *BnaATG8s* in the response of rapeseed plants to low phosphate stress, we analyzed their transcriptional expression under low phosphate conditions (Figure 3D). The results showed that under low phosphorus stress, a total of seven DEGs were identified, and their expression abundance showed an increasing trend, among which, *BnaC1.ATG8F*, *BnaA3.ATG8H**,* and *BnaC3.ATG8H* were significantly up-regulated in the roots.

### 2.7. Gene Co-Expression Network Analysis of the Core BnaATG8 Members

In the allopolyploid *B. napus*, the phenomenon of multiple gene copies was widespread. Therefore, the identification of core genes from multiple copies is the key to understanding the molecular mechanism of rapeseed. According to the transcriptional response of the *BnaATG8* gene family to various nutrient deficiencies and salt stress, we found that *BnaATG8s* had the most obvious response to nitrate starvation and salt stress (Figure 3C,D). Therefore, in order to identify the core genes, we selected the DEGs with high expression abundances based on these two transcriptomic data to construct two gene co-expression networks (Figure 3E,F). As shown in the co-expression network, *BnaA8.ATG8F* was identified as the core gene in the network.

### 2.8. BnaA8.ATG8F May Be Indispensable for The Resistance to Salt Stress and Low Nitrogen Stress

Next, in order to verify the function of *BnaA8.ATGF* under various stresses, we constructed the recombinant overexpression plasmid and obtained homozygous transgenic plants. Based on the transcriptomic analysis mentioned above, we further studied the response of transgenic plants to salt stress, low Pi stress, and low nitrate stress. The results showed that there was almost no difference between the wild type and the transgenic plants under low Pi conditions (Appendix A), whereas they showed different growth performances under salt stress and low nitrate conditions (Figure 4, Figure 5 and Figure 6). Because the transgene was driven by the cauliflower mosaic virus 35S promoter, we expected that the overexpression of *BnaA8.ATG8F* would enhance salt and nitrate deficiency tolerance. However, under the 200 mM NaCl or 0.3 mM NO_3_^−^ hydroponic condition, the transgenic plants appeared to have worse tolerance (Figure 4 and Figure 6). Subsequently, we determined the expression of *BnaA8.ATG8F* in wild-type and transgenic plants. Interestingly, the transcription level of *BnaA8.ATG8F* in the transgenic plants was downregulated to about 30% of those in wild-type. These findings suggested that the overexpression of *BnaA8.ATG8F* might lead to the occurrence of co-inhibitory events [30]. Therefore, we concluded that inhibition of *BnaA8.ATG8F* expression attenuates salt stress and low nitrate tolerance.

### 2.9. Effects of BnaA8.ATG8F on Morpho-Physiological and Mineral Nutrients in Rapeseed under Salt Stress

Subsequently, in order to further explore the role of *bnaa8.atg8f* in rapeseed salt tolerance, we studied the phenotypic and physiological changes of transgenic plants after salt treatment. After a 5-d exposure to salinity, the young leaves of *bnaa8.atg8f* plants were curved and wilting and showed obvious chlorosis, while those of WT were less severely affected. The roots of *BnaA8.ATG8F* also showed significant inhibition of growth. In addition, under salt treatment, the dry weight of shoots and roots was significantly lower in *bnaa8.atg8f* than in wild type (Figure 4). As the final product of membrane lipid peroxidation, MDA is one of the important indexes to measure cell membrane damage. The MDA concentration in *bnaa8.atg8f* was significantly higher than that in the wild type. In contrast, the wild type contained a higher concentration of proline. We further carried out a detailed examination of the roots. After 5 d of 200 mM NaCl treatment, the wild type had relatively better RSA than transgenic lines, including larger total root length, root surface area and volume, and more root tip numbers.

We next examined the ionomes of the WT and bnaa8.atg8f in the NaCl treatment using ICP-MS. The total whole-plant Na^+^ concentration was significantly higher in wild type than in bnaa8.atg8f, and WT plants also accumulated more Na^+^ (Figure 5A,B). Specifically, the Na^+^ concentration in shoots of bnaa8.atg8f was significantly higher than that of the wild type, but there was no significant difference between the two in roots (Figure 5C,D). The ratio of shoot/root Na^+^ accumulation in bnaa8.atg8f was almost three times that in WT (Figure 5I). In addition, there was no significant difference in K^+^ concentration and content between bnaa8.atg8f and wild type (Figure 5E,F). Notably, we observed lower root K^+^ concentration and content in bnaa8.atg8f (Figure 5G,H). Studies have shown that the ratio of Na^+^/K^+^ in cell solute determines the ability of cells to resist salt stress [31]. The results showed that the transgenic plants had higher Na^+^/K^+^ ratios in roots, shoots, and whole plants (Figure 5J–L). Therefore, a higher Na^+^ translocation coefficient seems to be the main reason for the difference in salt tolerance between wild type and bnaa8.atg8f.

We also examined other ion concentrations and found that Fe^2+^ and Zn^2+^ within the roots and shoots were both significantly higher in bnaa8.atg8f than in wild type. Mn^2+^ differed only in the roots and had a similar trend to Fe^2+^ and Zn^2+^. No differences in the concentrations of Cu^2+^, Ca^2+,^ and Mg^2+^ were observed between wild-type and bnaa8.atg8f plants (Appendix A).

To further prove that BnaA8.ATG8F can enhance salt tolerance through the root-to-shoot Na^+^ translocation pathway we compared the expression patterns of high-affinity potassium transporter (HKT) in bnaa8.atg8f and wild type. HKTs can reduce the transport of Na^+^ from roots to shoots by unloading Na^+^ in the roots [32]. In previous studies, we found that BnaC2.HKT1 plays a core role in salt stress resistance through long-distance transport in rapeseed. Therefore, we specifically investigated the expression of this gene [33]. The results showed that under normal conditions, there was almost no difference in the expression level of BnaC2.HKT1 between wild type and bnaa8.atg8f. Under salt stress, down-regulation of BnaA8.ATG8F expression led to more Na^+^ translocation to shoots, which reduced plant tolerance to salt (Figure 4D).

### 2.10. BnaA8.ATG8F Plays a Key Role in Rapeseed Resistance to Low Nitrate Stress

In order to further study the role of *BnaA8.ATG8F* in rapeseed tolerance to nitrate deficiency, we continued to explore the differences in response to low nitrate stress between wild type and *bnaa8.atg8f*. We found that under normal conditions (6 mmol NO_3_^−^), there was no significant difference in growth performance between bnaa8.atg8f and wild type. Under long-term low nitrate conditions, the growth of the bnaa8.atg8f shoots was inhibited, indicated by smaller leaves (Figure 6A). In addition, bnaa8.atg8f showed significantly reduced chlorophyll biosynthesis and excessive accumulation of anthocyanins under limiting nitrate, in which older leaves seemed to have more obvious symptoms of nitrate deficiency (Figure 6B–D).

Subsequently, the total N in plant tissues was measured. The results showed that the transgenic plants and wild type were significantly reduced under low N treatment compared with normal N treatment. Specifically, the total nitrogen concentration of shoot WT and transgenic plants decreased by 48.3% and 66.1%, respectively (Figure 6E). In the older leaves, they dropped 36.3% and 56.9% (Figure 6G). For the nitrogen concentrations in the roots, the wild type decreased by 38.1%, while transgenic plants decreased by 36.5% (Figure 6F). In addition, after the nitrogen starvation treatment, the total nitrogen accumulation of transgenic plants decreased by 73.9%, significantly higher than that of the wild type, which decreased by 62.1% (Figure 6I). The translocation ratio of transgenic plants decreased by 43.1%, which was significantly higher than that of wild type (31.0%) (Figure 6H).

To further explore the possible roles of *BnaA8.ATG8F* in nitrogen uptake, transport, and assimilation, we first determined the distribution of nitrate nitrogen in tissues under high N and low N conditions. The results showed that under nitrogen limitation, *bnaa8.atg8f* accumulated less nitrate nitrogen in the shoots, but there was no significant difference between *bnaa8.atg8f* and wild type in the roots (Figure 6J–K).

The down-regulation of *BnaA8.ATG8F* resulted in changes in the concentrations of total nitrogen and nitrate nitrogen in plant tissues. We further studied the genes involved in nitrate uptake and transport. Due to the complexity of the rapeseed genome, each homologous gene has multiple copies. Therefore, we selected the core gene according to the previous study [34]. High-affinity NO_3_^−^ transporters play a key role in nitrogen uptake when plants are chronically nitrogen deficient. We focused on the main regulator, *BnaNRT2.1,* and its partner, *BnaNAR2.1* [35]. The result showed *BnaC8.NRT2.1a* and *BnaC2.NAR2.1* of *bnaa8.atg8f* were significantly up-regulated under low N (Figure 6N,O). In addition, we also analyzed the expressions of *BnaNRT1.7* in the old leaves and *BnaNRT1.8* in the roots, and they are reported to play a role in regulating nitrogen distribution in source and sink organs and in regulating root-to-shoot transport, respectively. Under low nitrogen conditions, the expression levels of these two genes in *bnaa8.atg8f* were significantly lower than in the wild type (Figure 6P,Q). In addition, we also found that GS activity in *bnaa8.atg8f* was slightly inhibited under low nitrogen (Figure 6L,M).

In order to further understand the extensive variation of *BnaA8.ATG8F* in different rapeseed genotypes, we analyzed its expression between them. The expression of *BnaA8.ATG8F* in the low-nitrogen tolerant genotype H211 was higher than in the low-nitrogen sensitive genotype L307 (Figure 7A,B). We presumed that overexpression of *BnaA8.ATG8F* would enhance the adaptability of rapeseed plants to low nitrogen.

## 3. Discussion

Previous studies have shown that the ubiquitin-like protein ATG8s are involved in autophagy and play critical roles in the regulation of plant development and abiotic stress responses, especially nutrient starvation [36]. However, there were few systematic studies on *ATG8* in *B. napus* before. In this study, a total of 29 full-length *BnaATG8s* were identified, and a detailed bioinformatics analysis was performed on them. Moreover, we analyzed the transcriptional response of *BnaATG8s* under different nutrient stress conditions. Subsequently, we identified the core genes in the *BnaATG8* family through the co-expression network and characterized their roles under nitrogen deficiency and salt stress through transgenic analysis.

### 3.1. Integrated Bioinformatics Analysis Provided a Comprehensive Insight into the Molecular Characteristics of BnaATG8s

Through the identification of genome-wide *ATG8s* in *A. thaliana* and *Brassica* crops, we found that *ATG8B* and *ATG8G* have been lost in *B. rapa, B. oleracea*, and *B. napus* (Appendix A). We speculated that this might be a gene deletion phenomenon caused by gene function redundancy. Phylogenetic analysis showed that the *ATG8* family could be divided into three groups. Clade I and Clade Ⅱ contained most members of the *ATG8* family, while Clade Ⅲ contained only *ATG8Hs* and *ATG8Is* (Figure 1B). Notably, the members of Clade Ⅲ had a longer evolutionary distance than the other genes, possibly because Clade Ⅲ was more similar to the animal *ATG8* homologs, while the other genes were closely related to fungi [37]. The structural diversity of exons and introns provided some evidence for their different functions, and even some homologous copies of genes had different gene structures (Appendix A). In addition, the conserved motif analysis showed that all *BnaATG8s* shared multiple motifs. Among them, the C-terminal glycine residue is the most distinctive feature. In general, the newly synthesized ATG8 is cleaved by the cysteine protease ATG4 to expose the C-terminal glycine residue, but the BnaATG8H and BnaATG8I proteins may be able to interact with the autophagosome membrane without ATG4 treatment [38,39], perhaps due to the lack of additional amino acid residues at the C-terminal following the glycine residue (Figure 1D). Interaction network projections showed that all ATG8s could interact with ATG7 and ATG3, which is considered a ubiquitin-like pathway. These results suggested that the ubiquitin-binding system involving ATG8s might be not only a common but also an essential pathway in regulating autophagy in plant cells. We further analyzed the CREs of the *BnaATG8* promoter regions, which were involved in many biological processes such as abiotic stress response and hormonal response. A large number of transcription factor binding sites were found in the promoter regions, which responded to plant signals and participated in the stress responses. In *A**. thaliana*, TGA9, which binds to the TGACG motif, has been shown to interact with ATG8A and ATG8E [40].

### 3.2. Transcriptional Analysis Revealed the Response of the BnaATG8 Family to Various Nutrient Stresses and Identified the Core Member BnaA8.ATG8F

In this study, several DEGs of *BnaATG8s* were identified under four nutrient stress conditions, among which more *BnaATG8s* were mainly induced by nitrate deficiency and salt stress (Figure 3). It seems that none of the *BnaATG8s* were differentially expressed under cadmium or ammonium toxicity. We hypothesized that *BnaATG8s* had different response patterns to different nutrient stresses. Interestingly, most of the genes were identified in the shoot, and their expression levels were up-regulated. Although the expression abundance of some genes was significantly decreased after treatment, this phenomenon only occurred in the roots. The result indicated that *BnaATG8s* have different expression patterns in the roots and shoots (Figure 3). However, *BnaATG8s* were found to be expressed both under normal growth conditions and under starvation or stress, which was consistent with previous studies. In *A**. thaliana* and rapeseed, *ATG8F* expression is up-regulated under nitrogen starvation stress [38,41]. Some *ATG8* expression levels were also up-regulated under other stress conditions.

Allotetraploid rapeseed has a complex genome, often with multiple copies of one homologous gene. Therefore, identifying core regulatory genes is the key factor in exploring the molecular mechanism underlying key agronomic traits in rapeseed. Based on differential expression and co-expression network analysis under salt stress and nitrogen starvation (Figure 3), we proposed that *BnaA8.ATG8F* might be the central gene in response to stress in the *ATG8* family of *B. napus*.

According to the *B. napus* transcriptome Information Resource (BnPIR), we find that *BnaA8.ATG8F* seems to have high expression in many tissues, such as leaves, stems, roots, and stems (Appendix A). In addition, we found that the *BnaA8.ATG8F* expression seemed to be higher in the older leaves than in other tissues, seemingly because autophagy occurs more frequently in older tissues (Appendix A). We also focused on the short-term response of *BnaA8.ATG8F* to abiotic stress and hormones (0–24 h). The results showed that the differential expression of *BnaA8.ATG8F* was induced by freezing, osmotic stress, and salt stress (Appendix A). Compared with the results of this study, the expression abundance of *BnaA8.ATG8F* increased significantly after only 3 h of salt treatment, which indicated that the gene changes were much earlier than the physiological changes.

### 3.3. Molecular Strategy Model of BnaA8.ATG8F Regulating Salt Stress Resistance and Nitrogen Limitation Adaptation in Allotetraploid Rapeseed

In this study, the transgene was driven by the CaMV 35S promoter. We expected that the overexpression of *BnaA8.ATG8F* would enhance plant tolerance. However, the expression level of transgenic plants was decreased rather than increased (Figure 4). We presumed the high expression of the *BnaA8.ATG8F* itself and the introduction of the same exogenous gene led to the occurrence of the co-inhibition event, and the CaMV 35s strong promoter often enhanced the degree of co-inhibition [30,42]. These findings seemed to explain why transgenic plants showed high sensitivity to nitrogen limitation.

Subsequently, we proposed a model based on physiology, genomics, ionomics, and transcriptional level studies to elucidate the molecular strategies of *BnaA8.ATG8F* in response to salt stress and nitrogen limitation. Under salt stress, down-regulation of the expression level of *BnaA8.ATG8F* led to a decrease in the expression abundance of the high-affinity potassium transporter *BnaHKT**1;1* (Figure 8). Down-regulated *BnaA8.ATG8F* seemed to attenuate nitrogen uptake and transport to varying degrees. Both *BnaNRT2.1* and *BnaNAR2.1* were strongly induced to maintain adaptation to low nitrogen. In addition, the increased expression level of *BnaNRT1.8* resulted in more N being unloaded in the roots, which affected the distribution of NO_3_^−^ in the shoots and roots, and weakened plant photosynthesis. Increased *BnaNRT1.7* expression enhanced the remobilization of old leaves to young leaves, resulting in less nitrogen accumulation in old leaves. Ultimately, reduced GS activity led to a reduction in the physiological process of nitrogen assimilation (Figure 8). Interestingly, we also found that the expression level of *BnaA8.ATG8F* was higher in the low-nitrogen tolerant genotype than in the low-nitrogen sensitive genotype, which to some extent, explained the influence of *BnaA8.ATG8F* on plant adaptation to low nitrogen (Figure 7). In conclusion, when the expression of *BnaA8.ATG8F* was decreased, and the tolerance of rapeseed plants to salt stress and nitrogen limitation was significantly weakened.

## 4. Materials and Methods

### 4.1. Identification and Gene Nomenclature of Autophagy Gene 8 (ATG8) Family in Brassica

In this study, the protein sequences of the newly published *B**. napus* genome (Westar) [20] were downloaded from the *B**. napus* pan-genome information resource (BnPIR) [43] (http://cbi.hzau.edu.cn/ (accessed on 2 February 2022) and the protein sequences of *A. thaliana*, *B. rapa* and *B. oleracea* were downloaded from the following databases: The Arabidopsis Information Resource (TAIR10, https://www.arabidopsis.org/ (accessed on 2 February 2022)) for *A. thaliana,* Brassica Database (BRAD) v. 2.0 [44] (http://brassicadb.org/brad/ (accessed on 2 February 2022).) for *B. oleracea* (JZS_V2.0) and *B. rapa* (Chiifu_V3.0). In order to obtain a complete set of *ATG8**s* of these, an integrated method was employed in this study. First, a preliminary search was performed using the key word ‘autophagy 8′ in the TAIR database to collect *ATG8s* from *A. thaliana*. We used the amino acid sequences of *Arabidopsis* ATG8s as queries to perform a BLASTp search against the genome databases of *B. rapa*, *B. oleracea**,* and *B. napus*. Then, we obtained the HMM profile of the ATG8 domains from Pfam (http://pfam.xfam.org/ (accessed on 2 February 2022)) and the hmmsearch tool embedded in HMMER3.0 (http://hmmer.org/download.html (accessed on 2 February 2022)) was used to search for the HMM profile with an expected e-value of 0.01. A self-BLAST of hits was first carried out to remove the redundancy, and the alternative splices were manually excluded. The remaining proteins were considered the putative *ATG8s*. All the obtained ATG8s were subsequently submitted to the corresponding species database for confirmation.

In this study, according to the method previously proposed, *ATG8s* in *Brassica* species were named according to the following criterion: abbreviation of species name (1 uppercase letter) + chromosome (followed by a period) + name of gene homologs in *A. thaliana* [45]. For example, *BnaA3.ATG8**F* represents an *AtATG8**F* homolog on the A3 chromosome of *B. napus*.

### 4.2. Chromosomal Localization and Syntenic Analysis of ATG8s in Brassica

The location of *ATG8s* on the *B**. napus* chromosomes was obtained from the genomic annotation in the BnPIR database [20]. Tandem duplicated genes were regarded as an array of more than 1 gene within a 100-kb genomic region [46]. The method of syntenic analysis refers to the PGDD database development [47]. First, BLASTP algorithm-based searches among *Brassica* species were used to identify potential homologous gene pairs (E < 1e^−10^). Afterward, MCScanX was used to identify and analyze syntenic homologous pairs as input [48]. The results were displayed using the TBtools software [49].

### 4.3. Analysis of Evolutionary Selection Pressure and Phylogeny Relationships

To further understand the selection pressure of *ATG**s* during evolution, we calculated the synonymous (Ks), non-synonymous (Ka) nucleotide substitution rates, and Ka/Ks based on CDS sequence alignments of *ATG8s* from *A**. thaliana* and *B**. napus*. Clustal W (http://www.clustal.org/clustal2/ (accessed on 2 February 2022)) was used for pairing analysis of CDS genes without a stop codon [50], and then the results were submitted to KaKs_calculator (https://sourceforge.net/projects/kakscalculator2/ (accessed on 2 February 2022)) to figure out the final result with the yn00 method [51,52].

In order to analyze the phylogenetic characteristics of ATG8s, the Clustal W program was used to compare the full-length sequences of the *ATG8* gene family in *B. napus*, *B. rapa*, *B. oleracea*, and Arabidopsis to evaluate their evolutionary relationships. MEGA X was used to construct a phylogenetic tree of the ATG8 proteins with 1000 bootstraps by the neighbor-joining (NJ) method [53].

### 4.4. Identification of Exon-Intron Structures, Conserved Motifs, and Putative Cis-Acting Regulatory Elements (CREs)

To identify the gene structure of the *ATG8* family, TBtools was used to display the gene structure containing UTR, intron, and exon based on the genome annotation of *B**. napus* from the BnPIR database [43].

To examine the structural differences among the ATG8 proteins in *A. thaliana* and *B. napus*, the protein sequences were submitted to MEME [54] (http://meme-suite.org/tools/meme (accessed on 2 February 2022)) to characterize the conserved motifs. All the default parameters were used except for the following: the optimum motif width was set to 6–50 bp, and the maximum number of motifs was set to 10. For each *BnaATG8*, a 1.5-kb genomic sequence upstream from the start codon (ATG) was downloaded from the BnPIR database. All of these sequences were subjected to the plantCARE database (http://bioinformatics.psb.ugent.be/webtools/plantcare/html/ (accessed on 2 February 2022)) to predict putative CREs. The distribution of the over-represented CREs along the promoters was displayed using TBtools software, and the CREs were illustrated with the worldcloud2 package of the R program.

### 4.5. Molecular Characterization of the ATG8 Proteins

In order to further understand the molecular characteristics of ATG8, ExPASy (http://www.expasy.org/tools/protparam.html 2 February 2022) [55] was used to identify the amino acid number, theoretical isoelectric point (pI), molecular weight (MW), grand average of hydropathy (GRAVY), aliphatic index (AI), and instability index (II) of the ATG8 proteins. WoLFPSORT (http://www.genscript.com/wolf-psort.html (accessed on 2 February 2022)) [56] and ChloroP v. 1.1 localization.

To characterize the transmembrane helices of the AtATG8s and BnaATG8s, we submitted their protein sequences to the TMHMM 2.0 (http://www.cbs.dtu.dk/services/TMHMM/ (accessed on 2 February 2022)) program [57]. Phosphorylation sites of BnaATG8s were predicted by NetPhos 3.1 online server (http://www.cbs.dtu.dk/services/NetPhos/ (accessed on 2 February 2022)) [58]. STRING (Search Tool for Recurring Instances of Neighboring Genes) (https://string-db.org (accessed on 2 February 2022).) web-server was used to display and predict the association networks of the ATG8s [59].

### 4.6. Transcriptional Responses of BnaATG8s and Functionality Analysis under Different Nutrient Stresses

To further understand the transcriptional response of *BnaATG8s* to various nutrient stresses, the *B. napus* seedlings were hydroponically cultured according to the method described by previous researchers [60,61]. Specifically, plump seeds were selected for germination for 7 d, and uniformly growing seedlings were transferred to black plastic containers containing 10 L Hoagland nutrient solution. The basic nutrient solution contains 1.0 mM KH_2_PO_4_, 5.0 mM KNO_3_, 5.0 mM Ca(NO_3_)_2_·4H_2_O, 2.0 mM MgSO_4_·7H_2_O, 0.050 mM EDTA-Fe, 9.0 µM MnCl_2_·4H_2_O, 0.80 µM ZnSO_4_·7H_2_O, 0.30 µM CuSO_4_·5H_2_O, 0.37 µM Na_2_MoO_4_·2H_2_O, and 46 µM H_3_BO_3_. The nutrient solution was refreshed every 3 d to maintain constant ion concentration. The growth conditions of rapeseed seedlings are as follows: the light intensity of 200 μmol m^−2^ s^−1^, the temperature of 25 °C daytime/22 °C night, the light period of 16 h photoperiod/8 h dark, and the relative humidity of 70%.

For nitrogen deficiency stress, 7 d-old *B. napus* seedlings after seed germination were first grown under 6.0 mM nitrate (NO_3_^−^) for 10 d and then were transferred to low nitrate (0.3 mM NO_3_^−^) for 3 d for sampling and phenotypic identification. As for the ammonium toxicity treatment, the 7 d-old *B. napus* seedlings after seed germination were hydroponically cultivated under 6.0 mM nitrate (NO_3_^−^) for 10 d and were then transferred to an N-free nutrient solution for 3 d. Finally, the samples were collected after 6 h of treatment with excessive ammonium (6.0 mM NH_4_^+^). Under phosphate starvation treatment, the 7 d-old uniform *B. napus* seedlings after seed germination were hydroponically grown under 250 μM KH_2_PO_4_ for 10 d and were then transferred to 5 μM KH_2_PO_4_ for 3 d. With regards to the potassium deficiency treatment, 7-d-old uniformly growing *B. napus* seedlings after seed germination were first grown under high potassium (6.0 mM K^+^) for 10 d and then transferred to low potassium (0.03 mM K^+^) for 3 d for sampling and phenotypic observation. Under salt stress, 7-d-old rapeseed seedlings with uniform growth after seed germination were cultured for 10 d and then transferred to 200 mM NaCl for 1 d until sampling. Under cadmium toxicity treatment, 7-d-old seedlings after seed germination were cultured in a normal nutrient solution for 10 d and then transferred to 10 µM CdCl_2_ for 12 h for sampling.

Total RNA was extracted from the tissue using TRIzol^®^ Reagent (Plant RNA Purification Reagent for plant tissue) according to the manufacturer’s instructions (Invitrogen, CA, USA), and genomic DNA was removed using DNase I (TaKara, Shiga, Japan). Then RNA quality was determined by 2100 Bioanalyser (Agilent) and quantified using the ND-2000 (NanoDrop Technologies). Only high-quality RNA sample (OD260/280 = 1.8~2.2, OD260/230 ≥ 2.0, RIN ≥ 6.5, 28S:18S ≥ 1.0, >1μg) was used to construct sequencing library. RNA-seq transcriptome library was prepared following the TruSeqTM RNA sample preparation Kit from Illumina (San Diego, CA, USA) using 1μg of total RNA. Messenger RNA was first isolated according to the polyA selection method by oligo(dT) beads and then fragmented by fragmentation buffer. Secondly, double-stranded cDNA was synthesized using a SuperScript double-stranded cDNA synthesis kit (Invitrogen, CA) with random hexamer primers (Illumina). Then the synthesized cDNA was subjected to end-repair, phosphorylation, and ‘A’ base addition according to Illumina’s library construction protocol. Libraries were size selected for cDNA target fragments of 300 bp on 2% Low Range Ultra Agarose followed by PCR amplified using Phusion DNA polymerase (NEB) for 15 PCR cycles. After being quantified by TBS380, the paired-end RNA-sequencing library was sequenced with the Illumina HiSeq xten/NovaSeq 6000 sequencer (2 × 150 bp read length).

### 4.7. Differential Expression and Gene Co-Expression Network Analysis

To identify differentially expressed genes (DEGs) between 2 different samples in a variety of transcriptomes, the expression level of each transcript was calculated according to the transcripts per million reads (TPM) method. RSEM (http://deweylab.biostat.wisc.edu/rsem/ (accessed on 2 February 2022)) [62] was used to quantify gene abundances. Essentially, differential expression analysis was performed using the DESeq2 [63] with *p*-values ≤ 0.05, the genes with *p*-values <= 0.05 and log_2_FC| > 1 (DESeq2) were considered to be DEGs.

Gene co-expression networks were constructed to understand the expression trends among genes and identify core genes, which link most neighboring genes and participate in response to abiotic stress. For each pair of genes, the thresholds for Pearson correlation values were set according to the default parameters (http://plantgrn.noble.org/DeGNServer/Analysis.jsp (accessed on 2 February 2022)) and displayed using Cytoscape software (http://www.cytoscape.org/ (accessed on 2 February 2022)) [64].

### 4.8. Construction of Recombinant Vector and Acquisition of Transgenic Rapeseed

The rapeseed transformation vector pBWA(V)HS-gfp-3xflag (from Wuhan, China BioRun Biosciences Co, Ltd.) contained hygromycin as a selection marker. *BnaA8.ATG8F* (*BnaA08G0093200WE*) was cloned from “Westar” leaf cDNA. The *BnaA8.ATG8F CDS* (without the stop codon) was fused to the cauliflower mosaic virus 35S (CaMV 35S) vector by homologous recombinase (BioRun, Seamless Cloning Kit, Wuhan, China).

The plasmid constructs were then transferred into GV3101 and were subsequently used to transform ‘Westar 10′ as described previously [65]. Although we used the overexpression vector, the expression of transgenic lines did not increase but decreased, which may be due to the occurrence of co-inhibition events [30]. Therefore, these transgenic materials are considered loss-of-function mutants.

### 4.9. Determination of Osmoregulatory Substances, Chlorophyll, Anthocyanin, Nitrate Nitrogen, Total Nitrogen Concentration, and Enzyme Activities

Proline concentrations were determined spectrophotometrically at 520 nm using the ninhydrin method. Malondialdehyde (MDA) was extracted using thiobarbituric acid (TBA), and the concentration was determined by spectrophotometry at 532 nm and 600 nm wavelengths. Total chlorophyll was extracted using 80% isopropyl alcohol (*v/v*), and the concentrations of the purified extracts were then determined spectrophotometrically at 663.2, 646.8, and 470 nm, respectively. The concentration of anthocyanins was determined by an improved differential pH method [66]. The nitrate-nitrogen concentrations were determined spectrophotometrically at 410 nm, according to the method used by Patterson et al. [67]. The nitrogen concentration of plants was determined by the micro-Kjeldahl method with sulfuric acid-hydrogen peroxide digestion. The glutamine synthetase (GS) activity was assayed with the method reported by Wang et al. [68].

### 4.10. Ionomic Analysis

To determine ion contents, tissues sampled from the plants were over-dried at 65 °C to constant weight. To estimate cation concentrations (K^+^, Ca^2+^, Na^+^, Mg^2+^, Fe^2+^, Cu^2+^, Mn^2+^, and Zn^2+^), the samples were ground to a fine powder and incubated with an acid mixture of HNO_3_ and HClO_4_ (4:1, *v*/*v*) at 200 °C until complete digestion. To quantify the ion concentrations, diluted samples of the supernatant were subjected to inductively coupled plasma mass spectrometry (ICP-MS, NexIONTM 350X, PerkinElmer, Waltham, MA, USA)

### 4.11. Quantitative Reverse-Transcription PCR (RT-qPCR) Assays

After the treatment of RNase-free DNase I with RNA samples, the total RNA was used as the template for complementary DNA (cDNA) synthesis with the Takara PrimeScript™ RT reagent Kit with gDNA Eraser (Perfect Real Time). The qRT-PCR assays for the detection of relative gene expression were performed using SYBR^®^ Premix Ex Taq™ II (TaKaRa, Shiga, Japan) under an Applied Biosystems StepOne™ Plus Real-time PCR System (Thermo Fisher Scientific, Waltham, MA, USA). The thermal cycles were as follows: 95 °C for 3 min, followed by 40 cycles of 95 °C for 10 s, then 60 °C for 30 s. Melt curve analysis was conducted to ensure the primer gene-specificity as follows: 95 °C for 15 s, 60 °C for 1 min, 60–95 °C for 15 s (+0.3 °C per cycle). Expression data were normalized using the public reference genes *BnaEF1-α* and *BnaTublin**,* and relative gene expression was calculated with the 2^−ΔΔCT^ method. The gene-specific primers of genes for the RT-qPCR assays are listed in Appendix A.

### 4.12. Statistical Data Analysis

Statistical data were presented as mean ± SD (*n* = 3). Comparisons among different treatments were performed using Student’s *t*-test or 1-way ANOVA. *p* < 0.05 was considered statistically significant. *Pearson* correlation and statistical analysis were carried out using the GraphPad Prism 8 software.

## Figures and Tables

**Figure 1 ijms-23-11318-f001:**
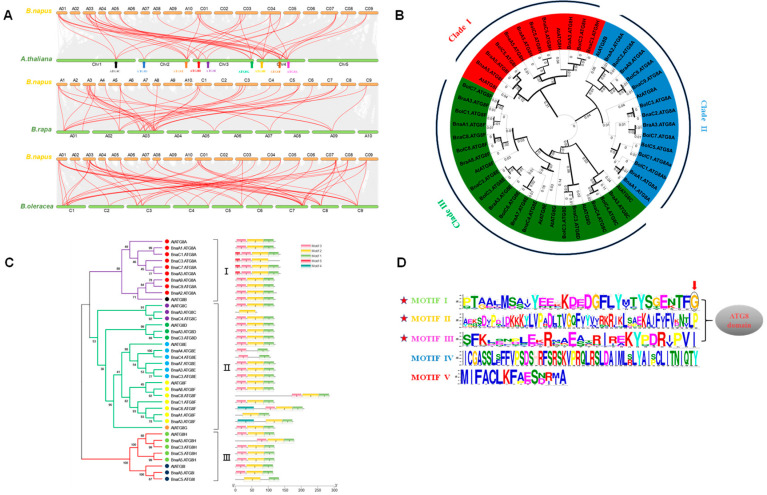
Identification and bioinformatics analysis of (*Autophagy-related gene 8*) *ATG8* Gene family in *Brassica napus*. (**A**) Syntenic analysis of the *ATG8* family genes in *A**. thaliana*, *B. rapa*, *B. oleracea*, and *B. napus* (**B**) Phylogeny analysis of the *ATG8* genes in *A**. thaliana* and *Brassica* crops. (**C**,**D**) Identification and characterization of the conserved motifs in the ATG8 proteins in *A**. thaliana* and *B**. napus*. Molecular identification (**C**) and sequence characterization (**D**) of the conserved motifs in the ATG8 proteins in *A. thaliana* and *B. napus.* L. In C, the boxes with different colors indicate different conserved motifs (motifs 1–5), and black lines represent the ATG8 protein regions without detected motifs. In D, the larger the fonts, the more conserved the motifs. Among them, the tagged motifs were identified as the ATG domains.

**Figure 2 ijms-23-11318-f002:**
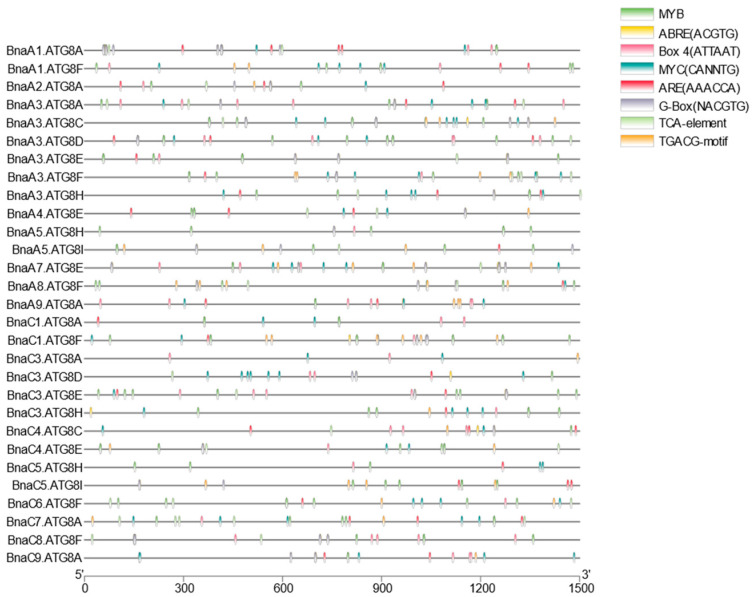
Identification of the putative *cis*-acting regulatory elements (CREs) of the 1.5-kb genomic sequences upstream of the start codon (ATG) of the *ATG8* family genes in *B. napus*. Genomic distribution and relative abundance of the eight most CREs in the *BnaATG8* promoters. Different CREs are indicated with different colors.

**Figure 3 ijms-23-11318-f003:**
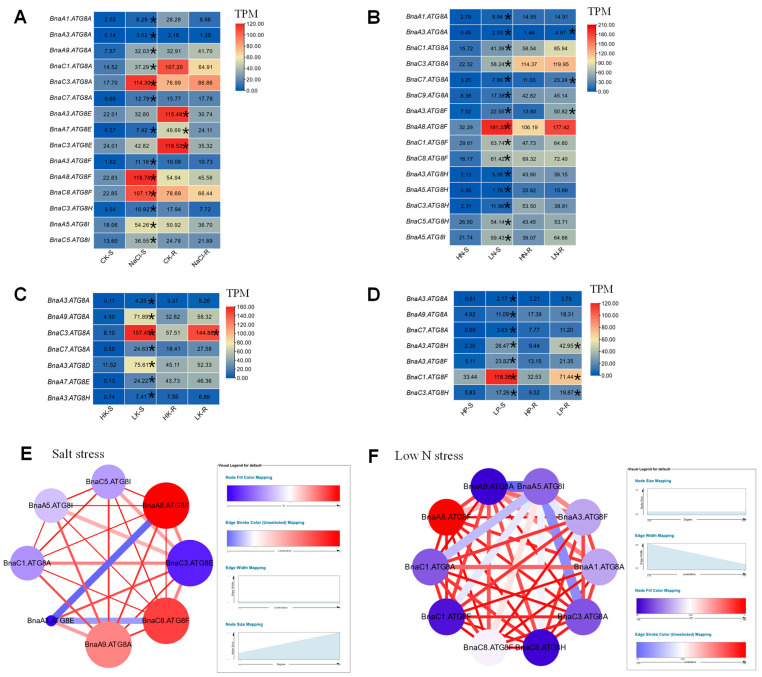
Transcriptional profiling of the differential expression of the *ATG8**s* in *B**. napus* under various nutrient stresses. (**A**–**D**) Differential expression profiling under salt stress (**A**), low nitrate stress (**B**), low potassium stress (**C**), and low phosphorus stress (**D**). (**E**,**F**) The gene co-expression network diagram involving *ATG8s* under salt and low nitrate stress. The shoots and roots were individually sampled, and each sample includes three independent biological replicates. The DEGs with the *p*-value <= 0.05 and |log_2_FC| > 1. Deleted genes with correlation values > 0.7 were used as the threshold for screening interactions between genes. The DEGs with higher expression are indicated with asterisks.

**Figure 4 ijms-23-11318-f004:**
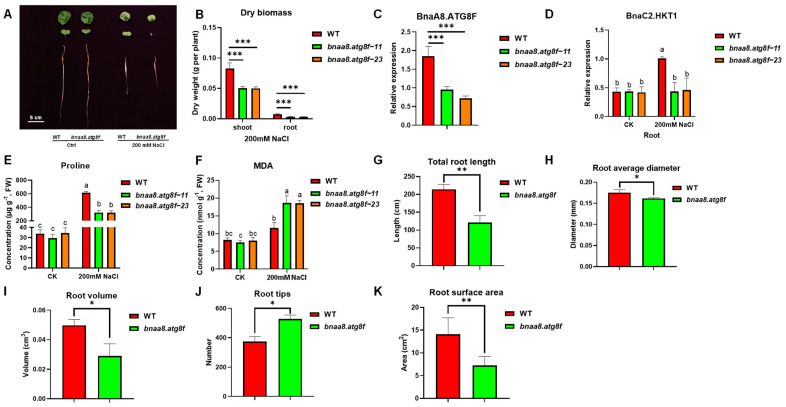
Downregulation of *BnaA8.ATG8F* reduced tolerance to salt stress. (**A**) Representative images of leaves and roots of the WT and *bnaa8.atg8f* after 5 d treatment with 200 mM NaCl. Ctrl, control. (**B**) Root and shoot dry weight. (**C**,**D**) Relative expression of (**C**) *BnaA8.ATG8F* in the plants and (**D**) *BnaC2.HKT1* in the roots. Expression is relative to that of *BnaEF1-α* and *BnaTublin*. (**E**,**F**) concentrations in younger leaves of (**E**) proline and (**F**) malondialdehyde (MDA). Data are means (±SD), *n* = 3. Different letters indicate significant differences as determined using a one-way ANOVA followed by Tukey’s HSD test (*p* < 0.05) and (**G**) total root length, (**H**) root average diameter, (**I**) root volume, (**J**) root tips, (**K**) root surface area. Significant differences were determined using Student’s *t*-test: * *p* < 0.05; ** *p* < 0.01; *** *p* < 0.001.

**Figure 5 ijms-23-11318-f005:**
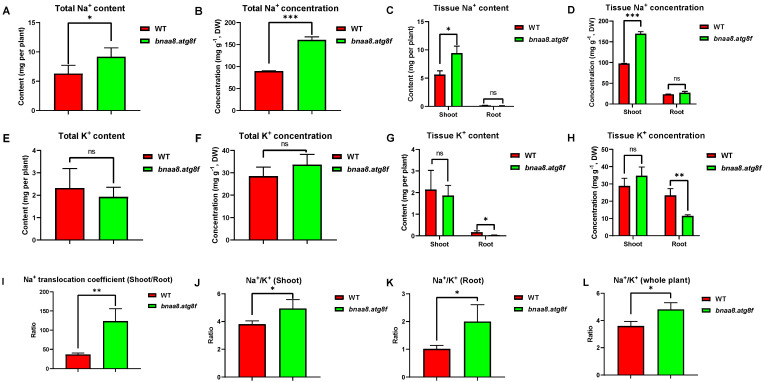
Sodium (Na) and potassium (K) ion profiling of the wild type and *bnaa8.atg8f* trangenic plants grown under salinity conditions. Following germination, uniform seedlings were selected and grown hydroponically under NaCl-free conditions for 10 d and then transferred to 200 mM NaCl for 5 d. (**A**) Total plant Na^+^ contents, (**B**) total plant Na^+^ concentrations, (**C**) shoot and root Na^+^ contents, (**D**) shoot and root Na^+^ concentrations. (**E**) total plant K^+^ contents, (**F**) total plant K^+^ concentrations, (**G**) shoot and root K^+^ contents, (**H**) shoot and root K^+^ concentrations, (**I**) Na^+^ translocation coefficient (root content/shoot content), (**J**–**L**) Na^+^/K^+^ ratios of (**J**) the shoots, (**K**) roots, and (**L**) the whole plant. Data are means (±SD), *n* = 3. Significant differences were determined using Student’ s *t*-test: *, *p* < 0.05; **, *p* < 0.01; ***, *p* < 0.001; ns, not significant.

**Figure 6 ijms-23-11318-f006:**
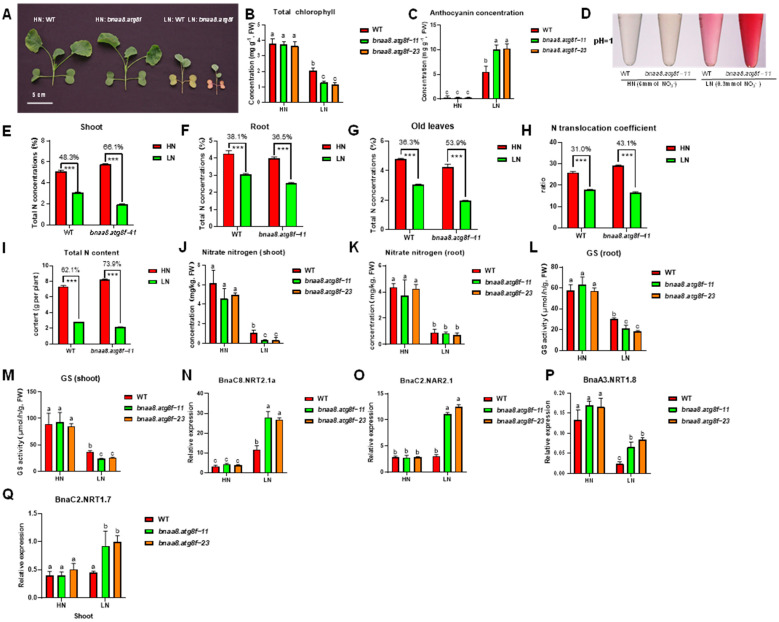
Down-regulation of *BnaA8.ATG8F* reduced adaption to nitrogen limitation. (**A**) Representative images of leaves of the wild type and *bnaa8.atg8f* after 10 d treatment with HN, 6 mM NO_3_^−^ and LN, 0.3 mM NO_3_^−^. (**B**) Chlorophyll concentrations in young leaves (**C**) Anthocyanin concentrations in the shoots. (**D**) Representative images of anthocyanin extract at pH = 1.0. (**E**–**G**) Total N concentrations in the (**E**) shoots, (**F**) roots, and (**G**) old leaves. (**H**) N translocation coefficient (shoot/root). (**I**) Total N content in the whole plants. (**J**) Nitrate N in the shoots. (**K**) Nitrate N in the roots. (**L**) GS activity in the roots. (**M**) GS activity in the roots. (**N**–**Q**) Relative expression of (**N**) *BnaC8.NRT2.1a* in the roots, (**O**) *BnaC2.NAR2.1* in the roots, (**P**) *BnaA3.NRT1.8* in the roots, (**Q**) *BnaC2.NRT1.7*. in the old leaves between wild type and *bnaa8.atg8f*. Significant differences were determined using either Student’s *t*-test (***, *p* < 0.001) or one-way ANOVA followed by Tukey’s HSD test (*p* < 0.05).

**Figure 7 ijms-23-11318-f007:**
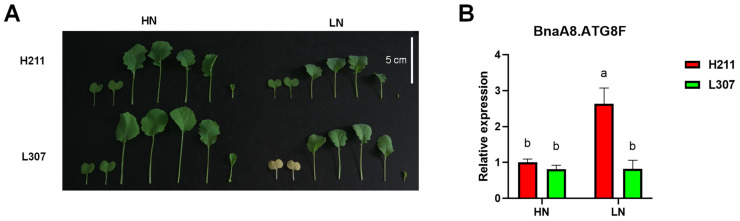
Natural variation in rapeseed resistance to low nitrate of resistant and sensitive genotypes. (**A**) Representative images of leaves of the low nitrogen tolerance genotype H211 and the low-nitrogen sensitive genotype L307 after 10 d treatment with HN, 6 mM NO_3_^−^ and LN, 0.3 mM NO_3_^−^. (**B**) Relative expression of *BnaA8ATG8F* between H211 and L307. Significant differences were determined using a one-way ANOVA followed by Tukey’s HSD test (*p* < 0.05).

**Figure 8 ijms-23-11318-f008:**
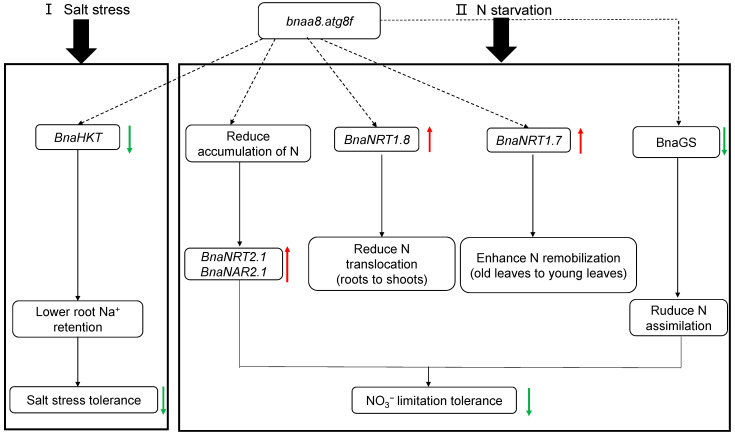
A proposed strategy model for the *BnaA8.ATG8F*-mediated regulation of nitrogen limitation adaptation and salt stress resistance in allotetraploid rapeseed. The dashed lines indicate potential or indirect regulation, and the red and green solid lines denote the up-regulation and down-regulation of gene expression by N starvation and salt stress.

## Data Availability

All the data and plant materials in relation to this work can be obtained through contacting with the corresponding author Dr. Yingpeng Hua (yingpenghua@zzu.edu.cn).

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
