# Peer review of "Genome-Wide Identification and Functional Characterization Reveals the Pivotal Roles of BnaA8.ATG8F in Salt Stress Tolerance and Nitrogen Limitation Adaptation in Allotetraploid Rapeseed"

_ijms, 2022, doi:10.3390/ijms231911318_

Round 1
Reviewer 1 Report
Through a genome-wide association search of the genes of the autophagy machinery, the authors found that the BnaA8.ATG8F gene has a role in tolerance to salt stress and nitrogen limitation.
This information seems interesting to me because it reflects the possibility of a selective gene expression related to the type of stress of the genes of the ATG8 family.
The manuscript could be divided into two parts, the first comprises the "classical" bioinformatic analysis, through which the authors deduce that the BnaA8.ATG8F gene is mostly involved in tolerance to salt stress and nitrogen limitation.
The second part comprises experiments designed to show that the BnaA8.ATG8F gene responds to salt stress and N deficiency. They employ gene silencing and overexpression and conclude that overexpression of this gene confers tolerance to salt stress and N deficiency.
Possible target genes in these response pathways to salt stress and nitrogen deficiency are analyzed and a model is proposed.
In general, the information provided seems interesting and useful for the field.
However, the writing is sloppy.
There are sections in italics and erroneous notes in the supplementary tables.
Some figures are left over like the supplementary S5.
The final version of the manuscript was not carefully reviewed, which seems serious to me.
The PDF document with notes is attached.
From the supplementary material:
Note: CDS, coding sequence; Ka, non-synonymous nucleotide substitution rate; Ks, synonymous nucleotide substitution rate. MOVE TO TABLE S3
Note: GRAVY, grand average of hydropathy; II, instability index; MW, molecular weight; pI, isoelectric point; PM, plasma membrane; TM, transmembrane; Vac, vacuole; Endo, endosome
MOVE TO TABLE S2

Author Response
Point-to-point response to Reviewer 1#
General comment: The article is interesting and detailed. It needs to be double-checked for technical and typographical errors.
Response: We appreciate for your warm work and thanks very much for your positive and constructive comments on our manuscript. Your comments are considerably valuable and very helpful for revising and improving our paper, particularly on the experimental design for proceeding the project in the future. We have studied your comments carefully and have made corrections in red with tracked changes which we hope meet with your approval. The point-by-point responses to your comments are also listed as below.
Comment 1: In the Materials and Methods section, the experimental setup states that the plants were grown in a hydroponic solution. In this study, Brassica plants were used as a model crop. Do the authors have information on the behavior of the same indicators when the plants are grown as a soil crop? Can the results obtained in this study be extrapolated to the conduct of Brassica plants grown in soil under the same stress conditions?
Response: Thank you very much for your kind suggestion.
As you have mentioned, the behavior of crops in soils may be more accurate for plant studies, particularly for those upland crop species, including Brassica napus. However, the heterogeneity of nutrient, water and other environmental factors in soils will bring about many unfavorable effect on the in-depth studies into the physiological and molecular mechanisms underlying key agronomic traits of crop species.
Indeed, hydroponic soil-free culture has been widely used in plant studies of many laboratories around the world due to its great feasibility. Moreover, the hydroponic culture system has been successfully used in the studies of Brassica napus in our research group. the The results obtained under the hydroponic culture system can be extrapolated to the conduct of Brassica plants grown in soil under the same stress conditions, which has been confirmed in our previous studies.
Some published papers about Brassica napus grown under the hydroponic culture system by our group are listed as follows:
Hua Y, Zhang D, Zhou T, He M, Ding G, Shi L, Xu F. Transcriptomics-assisted quantitative trait locus fine mapping for the rapid identification of a nodulin 26-like intrinsic protein gene regulating boron efficiency in allotetraploid rapeseed. Plant Cell Environ. 2016 Jul;39(7):1601-18. doi: 10.1111/pce.12731.
Hua Y, Zhou T, Ding G, Yang Q, Shi L, Xu F. Physiological, genomic and transcriptional diversity in responses to boron deficiency in rapeseed genotypes. J Exp Bot. 2016 Oct;67(19):5769-5784. doi: 10.1093/jxb/erw342.
Zhang ZH, Zhou T, Tang TJ, Song HX, Guan CY, Huang JY, Hua YP. A multiomics approach reveals the pivotal role of subcellular reallocation in determining rapeseed resistance to cadmium toxicity. J Exp Bot. 2019 Oct 15;70(19):5437-5455. doi: 10.1093/jxb/erz295.
Zhou T, Yue CP, Liu Y, Zhang TY, Huang JY, Hua YP. Multiomics reveal pivotal roles of sodium translocation and compartmentation in regulating salinity resistance in allotetraploid rapeseed. J Exp Bot. 2021 Jul 28;72(15):5687-5708. doi: 10.1093/jxb/erab215.
Comment 2: On pages 9 and 10, the main text is in italics. Please fix it.
Response: Thank you very much for your kind suggestion. According to your advice, we have changed them to the correct style.
Comment 3: In line 342, there is a repetition of the phrase "the difference."
Response: Thank you very much for your kind suggestion.
We have deleted the repetition of the phrase "the difference." in the Results section 2.9 of the revised manuscript, which is also as follows: “Therefore, higher Na+ translocation coefficient seems to be the main reason for the difference in salt tolerance between WT and bnaa8.atg8f.”
Comment 4: On line 398, the sentence must begin with the capital letter "we focused on ..."
Response: Thank you very much for your kind suggestion.
We have corrected the mistakes in the Results section 2.10 of the revised manuscript, which is also as follows: “We focused on the main regulator BnaNRT2.1 and their partners BnaNAR2.1.”
Once again, special thanks for your valuable comment and kind suggestion.
Reviewer 2 Report
The article is interesting and detailed. It needs to be double-checked for technical and typographical errors.
In the Materials and Methods section, the experimental setup states that the plants were grown in a hydroponic solution. In this study, Brassica plants were used as a model crop. Do the authors have information on the behavior of the same indicators when the plants are grown as a soil crop? Can the results obtained in this study be extrapolated to the conduct of Brassica plants grown in soil under the same stress conditions?
On pages 9 and 10, the main text is in italics. Please fix it.
In line 342, there is a repetition of the phrase "the difference."
On line 398, the sentence must begin with the capital letter "we focused on ..."
Please, recheck the text.
Author Response
Point-to-point response to Reviewer 2#
General comment: Through a genome-wide association search of the genes of the autophagy machinery, the authors found that the BnaA8.ATG8F gene has a role in tolerance to salt stress and nitrogen limitation.
This information seems interesting to me because it reflects the possibility of a selective gene expression related to the type of stress of the genes of the ATG8 family. The manuscript could be divided into two parts, the first comprises the "classical" bioinformatic analysis, through which the authors deduce that the BnaA8.ATG8F gene is mostly involved in tolerance to salt stress and nitrogen limitation. The second part comprises experiments designed to show that the BnaA8.ATG8F gene responds to salt stress and N deficiency. They employ gene silencing and overexpression and conclude that overexpression of this gene confers tolerance to salt stress and N deficiency. Possible target genes in these response pathways to salt stress and nitrogen deficiency are analyzed and a model is proposed.
In general, the information provided seems interesting and useful for the field.
Response: We appreciate for your warm work and thanks very much for your positive and constructive comments on our manuscript. Your comments are considerably valuable and very helpful for revising and improving our paper, particularly on the experimental design for proceeding the project in the future. We have studied your comments carefully and have made corrections in red with tracked changes which we hope meet with your approval. The point-by-point responses to your comments are also listed as below.
Comment 1: There are sections in italics and erroneous notes in the supplementary tables.
Response: Thank you very much for your kind suggestion.
According to your advice, we have corrected these mistakes you mentioned.
The notes in Tables S2 and Tables S3 have been moved to the correct places.
Comment 2: Some figures are left over like the supplementary Figure S5.
Response: Thank you very much for your kind suggestion.
The Supplementary Figure S5 was mentioned in the Results section 2.6, which is as follows: “The results showed that almost all of the 9 ATG8s in Arabidopsis thaliana were highly expressed in cytoplasmic matrix and vacuole (Supplemental Fig. S5).” We double-checked the text carefully and corrected the mistakes.
Comment 3: Section 2.1. Genome-wide identification of ATG8s in Brassica species: Page2, line83-87: “In order to compare the evolutionary diversity of ATG8 gene among Brassica species……”: There are some problems such as unclear expression of sentence pattern and omission of space. It should be better to say: the Arabidopsis genome contains nine ATG8 genes.
Response: Thank you very much for your kind suggestion. We revised the manuscript and the details are as follows: “The Arabidopsis genome contains nine ATG8 genes including AtATG8A, AtATG8B, AtATG8C, AtATG8D, AtATG8E, AtATG8F, AtATG8G, AtATG8H, and AtATG8I. In order to further compare the evolutionary diversity of ATG8 genes among Brassica species, B. rapa, B. oleracea, and B. napus were identified by the method mentioned below.”
Comment 4: Section 2.2. Genomic distribution, expansion and syntenic analysis of BnaATG8s: Page3, line115: “ATG8s are located in seven chromosome blocks, including U, O, D, G, J, N and F (Supplemental Table S2)”: I COULD NOT FIND THIS INFORMATION IN TABLE S2.
Response: Thank you very much for your kind suggestion.
We are very sorry for our mistake, and these results should be listed in the Supplemental Table S3. We have corrected it in the Section 2.2 of Results in the revised manuscript, which is also as follows:
“Specifically, ATG8s are located in seven chromosome blocks, including U, O, D, G, J, N, and F (Supplemental Table S3).”
Comment 5: Section 2.3. Phylogenetic relationships and evolutionary selection pressure of the ATG8 proteins: Page3, line122: “In Arabidopsis, the ATG8 family members were mainly classify ed into three clades:”: Make spelling mistakes
Response: Thank you very much for your kind suggestion.
We removed the extra space in section 2.3, which is as follows: “In Arabidopsis, the ATG8 family members were mainly classified into three clades.”
Comment 6: Section 2.4. Molecular Characterization of BnaATG8s: Page 3, line145: “Supplemental Table S3”
Response: Thank you very much for your kind suggestion. We corrected it in the section 2.4 of the manuscript, which is as follows: “and the corresponding derived amino acid numbers ranged from 66 to 286 (Supplemental Table S2).”
Comment 7: Section 2.4. Molecular Characterization of BnaATG8s: Page 4, line155: “Supplemental Table S3”
Response: Thank you very much for your kind suggestion.
We corrected it in section 2.4, which is as follows: “…which indicates that there are great differences in protein stability in the BnaATG8 family (Supplemental Table S3)”
Comment 8: Section 2.4. Molecular Characterization of BnaATG8s: Page 4, line159: The corresponding chart needs to be added
Response: “Supplemental Table S2” is added to the corresponding position in section 2.4. Details are as follows:” The results indicated that except AtATG8E and AtATG8H and their homologues in rapeseed were located in the nucleus, the remaining proteins were predicted to be located in the cytoplasm (Supplemental Table S2).
Comment 9: Page 6, Fig 2: I suggest to remove this cartoon
Response: Thank you very much for your kind suggestion. This word cloud showing the enriched CREs was removed.
Comment 10: Page 8, Fig 3: CHANGE COLOR IT IS DIFFICULT TO READ
Response: Thank you very much for your kind suggestion. We changed it to a more acceptable color and showed the parameters in the figure. The changed figure is as follows
We also improved the captions. We added the following to the caption: “Deleted genes with correlation value > 0.7 was used as the threshold for screening interactions between genes”.
Comment 11: Section 2.9. Effects of BnaA8.ATG8F on morpho-physiological and mineral nutrients in rapeseed under salt stress: Page 9, line 309: PLEASE, EXPLAIN ORIGIN OF THE MUTANTS.
Response: Thank you very much for your kind suggestion. The origin of the mutants were described in Section 2.8 and explained the phenomenon based on previous research findings in section 3.3. The specific contents are summarized as follows: Because the transgene was driven by the cauliflower mosaic virus 35S promoter, we expected obtain overexpressed lines. Interestingly, we measured the expression levels of several transgenic lines, but found no increase in the amount of lines. Even in some transgenic plants the expression was downregulated to about 30% of those in wild-type. We hypothesized that the high expression of the gene itself and the introduction of the same exogenous gene led to the occurrence of the co-inhibition event, and the Camv35s strong promoter often enhanced the degree of co-inhibition. The relevant references are as follows:
Cheng, X.; Wang, Z., Overexpression of COL9, a CONSTANS-LIKE gene, delays flowering by reducing expression of CO and FT in Arabidopsis thaliana. The Plant journal : for Cell and Molecular Biology 2005, 43, (5), 758-768.
Napoli, C.; Lemieux, C.; Jorgensen, R., Introduction of a Chimeric Chalcone Synthase Geneinto Petunia Results in Reversible Co-Suppression of Homologous Genes in trans. The Plant Cell 1990, 2, (4), 279-289.
Comment 12: Section 2.9. Effects of BnaA8.ATG8F on morpho-physiological and mineral nutrients in rapeseed under salt stress: Page 9-10, line 329-358: CHANGE FONT STYLE
Response: Thank you very much for your kind suggestion. We changed it to the correct font style in section 2.9.
Comment 13: Section 2.9. Effects of BnaA8.ATG8F on morpho-physiological and mineral nutrients in rapeseed under salt stress: Page 10, line 357: “Under salt stress…...than transgenic plants”: THIS STAMENET IS NOT CLEAR
Response: Thank you very much for your kind suggestion. The corresponding position of the manuscript in Results section 2.9 was changed to follows: “Down-regulation of BnaA8.ATG8F expression led to more Na+ translocation to shoots, which reduced plant tolerance to salt”.
We have double-checked the manuscript carefully and corrected the unclear points and mistakes in the manuscript. Once again, special thanks for your valuable comment and kind suggestion.

Round 2
Reviewer 1 Report
I found that the manuscript has been improved according to the suggestions.
I found some missing spaces between words, please revised carefully.
Congratulations on your interesting work.